# Phylogeny of *Crepidotus applanatus* Look-Alikes Reveals a Convergent Morphology Evolution and a New Species *C. pini*

**DOI:** 10.3390/jof8050489

**Published:** 2022-05-07

**Authors:** Soňa Jančovičová, Katarína Adamčíková, Miroslav Caboň, Slavomír Adamčík

**Affiliations:** 1Department of Botany, Faculty of Natural Sciences, Comenius University in Bratislava, Révová 39, 81102 Bratislava, Slovakia; sona.jancovicova@uniba.sk; 2Department of Plant Pathology and Mycology, Institute of Forest Ecology, Slovak Academy of Sciences Zvolen, Akademická 2, 94901 Nitra, Slovakia; katarina.adamcikova@ife.sk; 3Laboratory of Molecular Ecology and Mycology, Institute of Botany, Plant Science and Biodiversity Center, Slovak Academy of Sciences, Dúbravská cesta 9, 84523 Bratislava, Slovakia; miroslav.cabon@savba.sk

**Keywords:** Agaricomycotina, *Crepidotus malachius*, morphology, wood inhabiting fungi, Slovakia

## Abstract

*Crepidotus applanatus* is known as a common wood inhabiting fungus widely distributed throughout the Northern Hemisphere. There have been contrasting opinions about the delimitation and taxonomic treatment of the similar species *C. malachius*. Our phylogeny did not support the close relationship of these two morphologically similar species and the grouping of collections labelled by both names within each phylogenetic species reflects unreliable species delimitations in the traditional literatures. Both species inhabit the wood of deciduous trees, and our morphological analysis identified the size of basidiospores as a significant difference between them. The collections from *Pinus sylvestris* are recognised as a new species, *C. pini* sp. nov., and its morphological identification requires a combination of both basidiospore and cheilocystidia characters.

## 1. Introduction

Fungi of the genus *Crepidotus* (Fr.) Staude (Crepidotaceae, Agaricales, Agaricomycotina) have small to medium-sized pileate basidiomata with a typically reduced or absent stipe, a lamellate hymenophore, and a brown spore print. They grow on various dead plant substrates—mainly on wood, less on soil, and exceptionally on fruiting bodies of other fungi. They are circumpolarly widespread and relatively common from tropical to hemiboreal regions [1,2,3,4,5]. Kirk et al. [6] reported about 200 species described worldwide before 2008 and 55 new species have been published more recently (https://www.mycobank.org, accessed 28 April 2022).

The morphological concept of the genus *Crepidotus* is well supported by molecular phylogeny [7,8,9], but its infrageneric classification is still defined on the basis of morphological features [5,10,11]. Following the latest European monograph of the genus *Crepidotus* by Consiglio & Setti [10], the section *Sphaerula* Hesler & A.H. Sm., serie *Applanatus* Consiglio & Setti is defined by verrucose globose to subglobose basidiospores and white or whitish hygrophanous, as well as smooth pileus. Four species with the above mentioned characteristics are recognised in Europe: *C. applanatus* (Pers.) P. Kumm.; *C. malachius* (Berk. & M.A. Curtis) Peck; *C. malachioides* Consiglio, Prydiuk & Setti; and *C. stenocystis* Pouzar [12]. Consiglio & Setti [10] treated *C. stenocystis* as synonymous with *C. malachius* var. *trichifer* Hesler & A.H. Sm.

The recent phylogenetic studies have confirmed that all four species form independent species lineages, but the morphological concept of *C. applanatus* and *C. malachius* is ambiguous (the clade of *C. malachius* contained specimens identified as *C. applanatus*) [12,13]. Some older morphology-based studies recognized these two species [2,4], while others synonymized them [1,14] or did not mention the name *C. malachius* at all [3,15,16].

This study focuses on the *Crepidotus* taxa that morphologically correspond to *C. applanatus* and *C. malachius*. We used morphological and molecular tools to verify the delimitation of these species. Phylogenetic analyses were applied to assess relationships within *Crepidotus* species that have globose and subglobose basidiospores.

## 2. Materials and Methods

### 2.1. Sampling

The examined materials included 61 *Crepidotus* specimens with globose to subglobose ornamented basidiospores originally identified as *C. applanatus*, *C. crocophyllus* (Berk.) Sacc., *C. ehrendorferi* Hauskn. & Krisai, *C. malachioides*, *C. malachius,* and *C. stenocystis*. All specimens are from the Herbarium of Comenius University in Bratislava, Faculty of Natural Sciences, Department of Botany, Slovakia (SLO). The examined materials with collection details are listed in Appendix A.

### 2.2. DNA Extraction, PCR and Sequencing

Total genomic DNA was extracted from dried material using the EZNA Fungal DNA Mini Kit (Omega Bio-Tek Inc., Norcross, GA, USA) following the manufacturer’s instruction. The internal transcribed spacer (nrITS) region of the ribosomal DNA was amplified with primer pairs ITS5-ITS4, or alternatively ITS5-ITS2 and ITS3-ITS4 [17,18]. Cycling conditions included an initial denaturation step at 95 °C for 15 min, followed by 30 cycles of a denaturation step at 95 °C for 30 s, annealing step at 56 °C for 30 s, and elongation step at 72 °C for 90 s. A final elongation was carried out at 72 °C for 10 min. The large subunit ribosomal DNA (nrLSU) was amplified using the primers LR0R and LR5 [19]. The initial denaturation was followed by 35 cycles, each comprising of a denaturation at 95 °C for 1 min, an annealing at 50 °C for 1 min, and an extension at 72 °C for 1 min completed by a final extension for 10 min at 72 °C. Amplification of the DNA was performed in a PCR reaction mix consisting of approximately 2 ng/μL of template DNA, forward and reverse primers (10 pmol/μL), 5× HOT FIREPol^®^ Blend Master Mix (Solis BioDyne, Tartu, Estonia), and molecular grade water added up to 20 μL. The target fragments were purified using a PCR Purification Kit (Qiagen, Hilden, Germany). Sequencing was performed at the SEQme sequencing company (Dobříš, Czech Republic).

### 2.3. Phylogenetic Analyses

The ITS sequences with 93% similarity or higher to any of the studied taxa and the LSU with 97.5% similarity were retrieved from GenBank (Appendix A). The ITS alignment is also supplemented by sequences published by Jančovičová et al. [20] and by sequences of selected European species with non-globose basidiospores retrieved from Unite (https://unite.ut.ee, accessed on 14 April 2022) and GenBank (www.ncbi.nlm.nih.gov/genbank, accessed on 14 April 2022). The LSU alignment is supported by sequences from Kumar et al. [21]. Raw sequence files were edited in Geneious version R10 [22]. Intra-individual polymorphic sites having more than one signal were marked with NC-IUPAC ambiguity codes. Several important samples and lineages retrieved from GenBank were represented only by one sequence, therefore the ITS and LSU alignments were analysed separately. The final alignments were aligned by MAFFT version 7 using the strategy E-INS-i [23] and were manually improved in Geneious version R10 [20]. The hypervariable regions were removed by gBlocks [24] using settings for less stringent selection. The final datasets were analysed with two different methods: Maximum Likelihood (ML) and Bayesian inference (BI). Maximum likelihood was computed at an IQ-tree web server [25] under GTR+I settings with 1000 ultrafast bootstrap replicates. The aligned fasta files were converted to the nexus format using Mesquite 3.61 [26] and further analysed using MrBayes 3.2.6. [27] on XSEDE at the Cipres Science Gateway [28]. Bayesian runs were computed independently twice with four MCMC chains for 10 million generations until the standard deviation of split frequencies fell below the 0.01 threshold. The convergence of runs was visually assessed using the trace function in Tracer 1.6 [29]. Three samples of the genus *Simocybe* were selected as an outgroup. The results were further edited with TreeGraph 2 [30].

### 2.4. Morphological Analyses

Our morphological observations were focused on the collections corresponding to the morphological profile of *C. applanatus* and *C. malachius,* which was defined by the lack of orange tint on the lamellae and pilei, not the hymeniderm pileipellis structure and without the predominant narrowly utriform cheilocystidia [10] (Appendix A).

The macromorphological characters were described based on fresh materials. The width of the pileus denotes a dimension from right to left; the pileus diameter was measured as the shortest distance between the attachment to the substrate and the pileus margin. References to colours follow the findings of Kornerup & Wanscher [31]. The decay stages of wood were classified according to Ripková et al. [32] and Olsson & Jonsson [33].

The micromorphological structures were examined from dried herbarium specimens under an Olympus BX41 light microscope using an oil immersion lens at a magnification of 1000×. All tissues were examined in a combination of 3% aqueous solution of KOH and aqueous Congo red. The colour of the structures was only observed in 3% aqueous solution of KOH. The drawings were made with an Olympus U-DA drawing attachment at a projection scale of 2000×. An Olympus Camedia C-5060 digital camera was used to take microscopic photos. The micromorphology was based on 20 (basidia, basidiola, and tramal cells) and 30 (basidiospores, cheilocystidia, terminal, and subterminal cells of pileipellis) measurements per specimen. Except for tramal cells, the range of microscopic characters is given as the minimum, maximum (in parentheses), average ± standard deviation, and average values. For tramal cells, only the range of minimum and maximum sizes is given. The statistical analyses were computed in R 3.6.2 (R Core Team, 2020, Vienna, Austria) and the results were visualized as box plots using the ggplot2 package [34]. The graphical outputs were further processed in CorelDRAW X5 (Corel Corporation, Ottawa, ON, Canada). The frequency of occurrence of the micromorphological structures (cheilocystidia, terminal cells of pileipellis) is related to the following values: 100% = always, 50.1–99% = mostly, 25.1–50% = often, 5.1–25% = sometimes, 0.1–5% = rarely, 0% = never. The descriptions of micromorphological characters were based only on sequenced specimens (Appendix A). The morphological terminology is adopted from Berger et al. [35] and Vellinga [36].

Abbreviations: L = number of lamellae reaching the stipe/the (closest) point of attachment to the substrate; l = number of lamellulae between each pair of lamellae; and Q = ratio of length and width of basidiospores, cheilocystidia, and terminal cells of pileipellis.

## 3. Results

### 3.1. Phylogenetic Analyses

The topologies of ITS and LSU trees (Figure 1) were similar but not consistent in all parts. All the included European members of the genus with globose to subglobose basidiospores were strongly supported by ITS. Three collections collected on pine stumps and with similar morphology as *C. applanatus* were sister to *C. stenocystis* and they are part of a clade that is sister to typical *C. applanatus* collections from the wood of deciduous trees. These pine inhabiting collections are described as a new species, *C. pini*. In the ITS tree, the material that is identified as *C. malachius* was clustered with *C. crocophyllus* and *C. ehrendorferi* (BI = 0.95, ML = 66). In the LSU tree, *C. malachius* was only clustered with *C. crocophyllus* and *C. globisporus* (BI = 1, ML = 33), and *C. ehrendorferi* was placed in a separate clade nested in a large lineage mainly with species without globose spores. In conclusion, *C. applanatus* and *C. malachius* were placed in unrelated phylogenetic lineages and the *C. applanatus* lineage contained additional look-alike species *C. pini*.

### 3.2. Morphological Analyses

After an initial morphological analysis, the characters of basidiospores and cheilocystidia were selected as having potential distinguishing values and were supported by additional observations. The basidiospore width and cheilocystidia length of 30 elements per collection were measured (Appendix A).

There were significant differences in the basidiospore width between *C. applanatus* and *C. malachius*, with no overlapping average values or medians (Figure 2). For *C. applanatus*, 75% of the basidiospore width variability was below 5.5 μm, and for *C. malachius*, 75% of this character variability was above 6 μm. The new species, *C. pini*, had an intermediate size and its basidiospore width did not overlap with the median and average values of either of the previous species, but its interquartile range overlapped towards its 25th percentile with *C. applanatus* and towards its 75th percentile with *C. malachius*.

The interquartile values of cheilocystidia length overlapped all three species (Figure 3). There was no significant difference between *C. applanatus* and *C. malachius*. *Crepidotus pini* had average values of all three sequenced collections that were higher than 40 μm and the highest value among the collections of both former species was 39.6 μm. Considering a relatively high range of cheilocystidia length values of all species within the individual collections, this character needs to be used with caution.

In conclusion, *C. applanatus* and *C. malachius* can be identified by basidiospore width, but the combination of the former character and cheilocystidia length is necessary to identify *C. pini*. Another feature to support the identification of *C. pini* can be the cheilocystidia length/width ratio. The cheilocystidia width of all species is similar, but in *C. pini* they are mostly narrowly clavate, and this type is less frequent in the other two species (Table 1). *Crepidotus pini* also differs from *C. applanatus* and *C. malachius* by incrustation of hyphae in pileipellis and by consistent presence of yellow tints on the pileus surface (never being white when young).

### 3.3. Taxonomy

*Crepidotus applanatus* (Pers.) P. Kumm., Der Führer in die Pilzkunde: 74, 1871. (Figure 4 and Figure 5A).

Basionym: *Agaricus applanatus* Pers., Observationes Mycologicae 1:8, 1796. 

Neotype: L 986.062-019 (from Scotland, on hardwood stump).

Description: Basidiomata pileate, sessile (laterally or dorsally attached to the substrate) or with a rudimentary lateral stipe, gregarious in sparse to dense groups, often imbricate. Pileus spathuliform, flabelliform, rounded flabelliform, rarely reniform; convex, plano-convex to applanate; flat, sometimes umbonate or depressed near the point of attachment; 5–40 mm wide, 5–50 mm in diameter; hygrophanous; margin usually involute, becoming inflexed; entire, undulate or lobate, translucently striate (up to one-fifth to one-half of the pileus diameter); surface of young and fresh basidiomata white, when mature becomes pale yellow (3A3) or pale orange (5B3), when dried sometimes becomes lighter (3A2–yellowish white); sericeous or velutinous, smooth; at the point of attachment white mycelial tomentum extending up to one-fifth of the pileus diameter. Stipe (if present) cylindrical, 2–7 × 2–4 mm, whitish, pubescent. Lamellae l = (3) 7(15); L = 12–26, 2–4 mm wide, ventricose, adnexed to decurrent; when young whitish to cream (4A3), when mature more orange and gray (5B3–pale orange, 5B4–grayish orange) to golden blond (5C4), the oldest coloured up to brown (5D6–oak brown or 5E7–linoleum brown); lamellae edges concolorous when young and paler (whitish) than lamellae sides when mature; entire or irregularly serrulate. Spore print brown (5E7–linoleum brown, 6E6–cacao brown). Context up to 2 mm thick, whitish to ivory (4B3), smell and taste indistinct.

Basidiospores (4.2)4.8–5.2–5.5(6.8) × (4.2)4.8–5.1–5.5(6.8) μm, Q = (0.94)0.99–1.00–1.02(1.10), globose, rarely subglobose, yellowish to yellowish brown, under light microscope with fine, isolated warts. Basidia 4-spored, rarely 2-spored, (16)19.8–23–26.1(32) × (6)6.5–7.1–7.7(9) μm, mostly clavate, rarely narrowly clavate, hyaline, thin-walled. Basidiola (11)13–16.2–19.4(24) × (4)5–5.8–6.6(7) μm, clavate, hyaline, thin-walled. Cheilocystidia (22)28.8–34.9–40.9(57) × (6)8.1–10–11.9(16) μm, mostly clavate, sometimes narrowly clavate, rarely utriform, ventricose, narrowly ventricose, narrowly cylindrical, forked, narrowly forked, lobate and narrowly lobate, obtuse at apex, hyaline, thin-walled. Pileipellis a cutis, (25)50–100(130) μm deep, composed of 5–15 layers of parallel, hyaline, thin-walled and non-incrusted hyphae; terminal cells (13)24.4–39.1–49.8(87) × (5)6–7.9–9.8(13) μm, repent or ascending to erect, often narrowly clavate, sometimes clavate, narrowly cylindrical and narrowly lageniform, rarely forked, cylindrical and narrowly ventricose, obtuse at apex; the first subterminal cells (17)28.6–42.3–56(77) × (3)5–6–7(9) μm, narrowly cylindrical or cylindrical. Trama of lamellae ± regular, hyphae 3–22 μm wide, cylindrical or widest in the middle or near septa, hyaline, thin-walled, without incrustations; trama of pileus without a gelatinous layer, hyphae 3–12 μm wide, in other characters similar to hyphae in lamellae, but some hyphae also dichotomously branched or anastomosed. Clamp connections present in all parts.

Ecology: Most of our collections of *C. applanatus* were from the wood of fallen trunks, fewer were from fallen branches, and one collection was from a standing trunk. The most common host tree was *Fagus sylvatica*, and less common was *Acer pseudoplatanus*, *Quercus* sp., and *Salix* sp.; that it grew on *Picea abies* is questionable. The fallen trunks and branches were usually 20 to 40 cm thick, in full contact with the soil and basidiomata grew mainly on their lateral side; basidiomata on a standing trunk grew to a height of about 150 cm. The substrates of the 3rd to 5th decay stages were often overgrown with mosses. The findings were from the months of July, September, and October. Most of our collections were from old-growth forests protected as nature reserves in Slovakia, few collections were from managed forests. Of the habitats, the species was most common in beech and fir-beech forests (*Fagus sylvatica* dominates the tree layer, along with admixed *Abies alba*, *Acer pseudoplatanus*, *Picea abies,* and others); one collection was from oak-hornbeam woods (tree layer is predominantly occupied by *Carpinus betulus*, *Quercus petraea,* and *Q. robur*); another one was from a former pasture having gradually been covered with wood trees of *Betula pendula*, *Populus tremula,* and *Salix caprea*; and one Norwegian collection was from a boreal *Picea abies* forest. The elevation of the stands was 60–900 m a.s.l.

*Crepidotus malachius* (Berk. & M. A. Curtis) Peck, Annual Report on the New York State Museum of Natural History 39: 71, 1886. (Figure 5B and Figure 6).

Basionym: *Agaricus malachius* Berk. & M.A. Curtis, Annals and Magazine of Natural History 4: 291, 1859. 

Holotype: ALT 5730 (from New England U.S.A.); isotype: FH-258654.

Description: Basidiomata pileate, sessile (laterally or dorsally attached to the substrate) or with a rudimentary lateral stipe, gregarious in sparse to dense groups, often imbricate. Pileus rounded flabelliform, flabelliform, rarely reniform; convex, plano-convex to applanate; flat, sometimes umbonate or depressed near the point of attachment; 8–75 mm wide, 10–50 mm in diameter; hygrophanous; margin usually involute, becoming inflexed, rarely straight; entire, in larger basidiomata often undulate, translucently striate (up to one-eighth to one-third of the pileus diameter); surface of young and fresh basidiomata white (watery white when soaked), when mature becomes yellowish gray (4B2–putty) or ivory (4B3), when dried sometimes more yellow (4A3–cream, 4A4–light yellow); sericeous or velutinous, smooth; at the point of attachment white mycelial tomentum extending up to one-eighth to one-quarter of the pileus diameter. Stipe (if present) cylindrical, 1–3 × 1–2 mm, whitish, pubescent. Lamellae l = 7–15; L = 14–28, 1–7 mm wide, ventricose, adnexed to decurrent; when young whitish to orange-white (5A2), when mature more gray and orange (5B2–orange-gray, 5B3–pale orange, 5C3–brownish orange) to golden blond (5C4) and up to brown (5D6–oak brown or 5E6–mustard); lamellae edges concolorous when young, paler (whitish) than the lamellae sides when mature; entire or irregularly serrulate. Spore print brown (5E6–mustard brown, 6E6–cacao brown). Context up to 2 mm thick, whitish to ivory (4B3), smell and taste indistinct.

Basidiospores (5)5.9–6.7–7.4(9.5) × (5)5.9–6.6–7.3(9.5) μm, Q = (0.95)0.98–1.01–1.04(1.15), globose, rarely subglobose, yellowish to yellowish brown, under light microscope with fine, isolated warts. Basidia 4-spored, rarely 2-spored, (18)24.7–28.5–32.3(41) × (7)7.8–8.7–9.7(12) μm, mostly clavate, rarely narrowly clavate, hyaline, thin-walled. Basidiola (13)17.2–20.9–24.6(30) × (4.5)6.3–7.7–9.1(11) μm, clavate, hyaline, thin-walled. Cheilocystidia (18)25.1–31.5–37.8(60) × (7)8.2–10.1–11.9(19) μm, mostly clavate, sometimes narrowly clavate and ventricose, rarely utriform, narrowly utriform, narrowly ventricose, cylindrical, narrowly cylindrical, forked, narrowly forked, lobate, pyriform and globose, obtuse at apex, hyaline, thin-walled. Pileipellis a cutis, (30)50–80(100) μm deep, composed of 6–12 layers of parallel, hyaline, thin-walled and non-incrusted hyphae; terminal cells (22)32.8–44.1–55.4(73) × (5)6.7–8.3–9.9(12) μm, repent or ascending to erect, often narrowly cylindrical and narrowly clavate, sometimes clavate, narrowly lageniform and narrowly ventricose, rarely cylindrical, ventricose and lageniform, obtuse at apex; the first subterminal cells (16)25.9–36.9–47.9(72) × (5)6.5–8.4–10.2(14) μm, narrowly cylindrical, cylindrical or broadly cylindrical. Trama of lamellae ± regular, hyphae 4–20 μm wide, cylindrical or widest in the middle or near septa, some anastomosed, hyaline, thin-walled, without incrustations; trama of pileus without a gelatinous layer, hyphae 4–23 μm wide, in other characters similar to hyphae in lamellae, but some hyphae also dichotomously branched. Clamp connections present in all parts.

Ecology: Most of our collections of *C. malachius* were from the wood of fallen trunks, fewer from stumps, and one collection was from a standing trunk and one from a fallen branch. The most common host tree was *Fagus sylvatica*, less common were *Acer campestre*, *Fraxinus excelsior,* and *Quercus* sp. The stumps and trunks were about 20 to 50 cm thick, and the branches were about 10 cm thick; fallen substrates were usually in full contact with the soil and basidiomata grew mainly on their lateral sides; basidiomata on a broken (still standing) trunk grew to a height of about 90 cm. The substrates of the 3rd to 5th decay stages were often overgrown with mosses. The findings were from the months of June to October. Most of our collections were from old-growth forests protected as nature reserves in Slovakia with few collections from managed forests. Of the habitats, the species was most common in beech and fir-beech forests (*Fagus sylvatica* dominates the tree layer, along with admixed *Abies alba*, *Acer pseudoplatanus*, *Fraxinus excelsior*, *Picea abies,* and others); few collections were from oak-hornbeam woods (tree layer is predominantly occupied by *Carpinus betulus*, *Quercus petraea,* and *Q. robur*); the collections from the Czech Republic were from the mixed forests of *Fagus sylvatica*, *Quercus petraea*, *Carpinus betulus,* and *Abies alba*; and one Russian collection was from a broad-leaved forest with *Tilia*, *Acer*, *Quercus,* and *Populus tremula* trees. The elevation of the stands was 90–850 m a.s.l.

*Crepidotus pini*, Jančovičová, sp. nov. (Figure 5C, Figure 7 and Figure 8).

MycoBank No.: MB843563.

Diagnosis: Basidiomata pileate, sessile, gregarious. Pileus rounded flabelliform, convex to applanate, 7–50 mm wide, 6–30 mm in diameter, hygrophanous; margin not or only when very moist translucently striate; surface grayish-yellow when young, brown when mature, smooth, when young pruinose, soon glabrous, sometimes with brown scales. Lamellae yellowish-white to grayish-yellow when young, brown when mature. Basidiospores are globose to subglobose, in average 5.8 × 5.7 μm, Qav. = 1.01, warted. Cheilocystidia mostly narrowly clavate, in average 43.3 × 9.8 μm. Pileipellis a cutis, the terminal cells mostly narrowly clavate. Clamp connections present in all parts. On the wood of decaying stumps of *Pinus sylvestris*.

Etymology: the epithet “pini” refers to the association with *Pinus* trees; the fungus was recorded on old decaying stumps of *Pinus sylvestris*.

Holotype: Slovakia, Záhorská nížina Lowland, Záhorie military district, Rohožník village, 48°27′41″ N, 17°08′52″ E, 210 m a.s.l.; about 100 years old forest of cultivated *Pinus sylvestris* and self-sown *Carpinus betulus*, *Quercus* and *Crataegus*; on wood at the stump base of a felled *Pinus sylvestris*, stump diam. ca. 40 cm, 4th decay stage, 12 September 2020, leg. S. Jančovičová (SLO 2579).

Description: Basidiomata pileate, sessile (even without a rudimentary stipe on the youngest basidiomata; laterally or dorsally attached to the substrate); growing in tight groups of more than five to nearly a hundred basidiomata (usually on one place on the stump), imbricate; or growing in more sparse groups of fewer than five basidiomata, sometimes single (often scattered in several places on the stump). Pileus mostly rounded flabelliform, sometimes flabelliform; convex, plano-convex to applanate; flat or umbonate, sometimes depressed near the point of attachment; 7–50 mm wide, 6–30 mm in diameter; hygrophanous; margin usually involute, becoming inflexed to reflexed, somewhat exceeding the lamellae; entire, in some (mostly in larger) basidiomata crenulated, crenate or undulate, not or only when very moist translucently striate; surface of young basidiomata grayish-yellow (4B4–champagne or 4B5–corn), with a distinctly lighter margin (4A4–light yellow), just a short time pruinose, soon glabrous, smooth, when drying with no or slightly lighter colour changes (4A4, 4B4, 4B5), mature and fresh basidiomata more brownish (5D5–clay, 5D6–oak brown, 5E6–mustard brown) to dark brown when very old (7F8), of the same colour or lighter when dried (4B4, 4B5); sometimes with fine sun brown (6D5) scales on the clay (5D5) background in older basidiomata; whitish mycelial tomentum at the point of attachment present on both sides of the pileus, strongly contrasting with the colour of pileus and lamellae, extending up to one-fifth to one-quarter of the pileus diameter. Lamellae l = (3)5–7; L = 9–22, 1–5 mm wide, ventricose, adnexed to decurrent; when young yellowish-white to grayish-yellow (4A2–yellowish white, 4A3–cream, 4B3–ivory, 4B4–champagne, 4B5–corn), when mature blond (5C4–golden blond to 5D4–dark blond), the oldest coloured up to brown (5E6–mustard brown or 5E7–linoleum brown) to dark brown (7F8); lamellae edges concolorous or paler (whitish) than lamellae sides; entire or irregularly serrulate. Spore print brown (6E5 to 6E6–cacao brown, 6F5 to 6F6–dark brown). Context up to 2 mm thick, grayish yellow (4B4–champagne) to dark blond (5D4), smell indistinct, taste indistinct or slightly like mushrooms or sour.

Basidiospores (4.8)5.3–5.8–6.3(7) × (4.8)5.2–5.7–6.2(7) μm, Q = (0.95)0.98–1.01–1.04(1.15), globose, sometimes subglobose, yellowish to yellowish-brown, under light microscope with fine, isolated warts. Basidia 4-spored, rarely 2-spored, (18)20.8–23.6–26.5(30) × (6)6.7–7.3–7.9(9.5) μm, mostly clavate, sometimes narrowly clavate, hyaline, thin-walled. Basidiola (9.5)14.1–17.4–20.7(26) × (4.5)5.5–6.4–7.3(9) μm, clavate, hyaline, thin-walled. Cheilocystidia (29)35.5–43.4–51.3(66) × (7)8.5–9.8–11.1(14) μm, mostly narrowly clavate, often clavate, sometimes narrowly utriform, rarely utriform, narrowly cylindrical, forked, narrowly forked, lobate and narrowly lobate, obtuse at apex, hyaline, thin-walled. Pileipellis a cutis, (50)70–100(120) μm deep, composed of 6–10 layers of hyaline or yellowish, thin-walled, incrusted or non-incrusted parallel hyphae; terminal cells (28)37.4–53.7–70(103) × (7)7.5–10.1–12.8(26) μm, repent or ascending to erect, mostly narrowly clavate, sometimes clavate, narrowly lageniform, narrowly utriform and narrowly cylindrical, and rarely utriform, cylindrical, ventricose, narrowly ventricose and forked, obtuse at apex, hyaline, some yellowish to yellowish-brown, thin-walled, non-incrusted; the first subterminal cells (10)19.4–31.7–44.1(73) × (6)7.2–9.8–12.4(17) μm, narrowly cylindrical, cylindrical or broadly cylindrical. Trama of lamellae ± regular, hyphae 4–20 μm wide, cylindrical or widest in the middle or near septa, hyaline, thin-walled, without incrustations, trama of pileus without a gelatinous layer, hyphae 4–18 μm wide, in other characters similar to hyphae in lamellae, but some hyphae also dichotomously branched. Clamp connections present in all parts.

Ecology: Our collections of *C. pini* were from 80 to 100 years old stumps of *Pinus sylvestris* in the 3rd to 5th decay stages. The stumps of felled trees were 30 to 50 cm thick and the basidiomata were formed at their base, on strongly decayed wood, and almost in contact with soil. Some stumps were overgrown with mosses (*Hypnum cupressiforme*, *Aulacomnium androgynum*) and some were also colonized by other basidiomycetes (*Mycena* sp., *Artomyces* sp.), ascomycetes (*Ascocoryne* sp.), or myxomycetes. The stumps were located in well-lit places, namely along the forest road, at the forest edge, or where the clearing was created by the felling of trees. The findings were from the months of August to October. We have found *C. pini* at three localities in the Záhorská nížina Lowland (north-western Slovakia), in forests with cultivated Scots pine (*Pinus sylvestris*), and at an elevation of 190 to 220 m a.s.l. The forests were between 80 and 100(110) years old and were reaching felling age.

Additional specimens examined: Slovakia, Záhorská nížina Lowland, Záhorie military district, Malacky town, 48°26′15″ N, 17°05′03″ E, 190 m a.s.l.; about 80 years old forest of cultivated with admixed *Quercus*; on wood at the stump base of a felled *Pinus sylvestris*, stump diam. ca. 50 cm, 4th decay stage, 30 September 2017, leg. S. Jančovičová (SLO 2550). Slovakia, Záhorská nížina Lowland, Záhorie military district, Rohožník village, 48°27′41″ N, 17°08′52″ E, 210 m a.s.l.; about 100 years old forest of cultivated *Pinus sylvestris* and self-sown *Carpinus betulus*, *Quercus,* and *Crataegus*; on wood at the stump base of a felled *Pinus sylvestris*, stump diam. ca. 40 cm, 4th decay stage, 1 October 2020, leg. S. Jančovičová (not.). Ibidem, the stump mechanically crushed, 4th to 5th decay stage, 26 September 2021 (SLO 2623). Ibidem, on wood at the stump base of a felled *Pinus sylvestris*, stump diam. ca. 35 cm, 4th decay stage, 3 October (not.). Ibidem, on wood at the stump base of a felled *Pinus sylvestris*, stump diam. ca. 50 cm, 4th to 5th decay stage, 10 October 2021 (SLO 2622). Slovakia, Záhorská nížina Lowland, Záhorie military district, Rohožník village, 48°27′18″ N, 17°07′39″ E, 220 m a.s.l.; about 100 years old forest of cultivated *Pinus sylvestris*; on wood at the stump base of a felled *Pinus sylvestris*, stump diam. ca. 40 cm, 3rd decay stage, 28 August 2021, leg. S. Jančovičová (SLO 2601). Ibidem, 19 September 2021 (SLO 2602).

## 4. Discussion

Our analyses confirmed that *C. applanatus* and *C. malachius* are morphologically distinct and phylogenetically unrelated taxa. These two species show similar field appearance and basidiospores shape, but they are not closely related. For example, *C. malachius* is more closely related to *C. crocophyllus*, the species with brown squamulose to fibrillose pileus. Members of serie *Applanatus,* which is defined by a white or whitish and hygrophanous pileus surface as well as globose to subglobose basidiospores, are not monophyletic and represent a morphotype that evolved convergently.

The nomenclature of both species defined in our phylogenetic study is assigned based on the basidiospore dimensions of the types reported by Consiglio & Setti [10], which agree with our basidiospore statistics (Figure 2). Basidiospore-based differences that were identified in our statistical analyses also correspond to the species concept presented by Hesler & Smith [2] and adopted in the most recent monographic studies of the genus [4,10,37] (Table 2). These two species had not been distinguished for a long time in the past, probably starting from Pilát [1]—who synonymized them in his monograph on European *Crepidotus* species. The species *C. malachius* was not mentioned in some monographic works on the genus *Crepidotus* in Europe, e.g., Watling & Gregory [38] focused on British taxa, Norstein [15] on Norwegian taxa, Senn-Irlet [3,39] on European taxa, Gonou-Zagou & Delivorias [40] on Greek taxa, or Pouzar [16] on Czech taxa.

Singer [41] treated *C. malachius* as a synomym of *C. nephrodes*. However, the phylogenetic studies clearly demonstrated that the latter species is a synonym of another species, *C. crocophyllus* [12,42].

Our phylogenetic study showed little variation in ITS or LSU regions among samples of *C. applanatus* that is distributed throughout the Northern Hemisphere, but there is a geographic clustering observed within *C. malachius* (Figure 1). This variation needs further analyses of additional loci, preferably including protein coding genes. If there are geographic variants or closely related taxa diversified by geographical distance, they may correspond to some varieties of either species. However, previously described varieties may also represent completely different taxa, for example *C. malachius* var. *trichifer* was synonymised with *C. stenocystis* [10].

The new species *C. pini* seems to have strict preference to conifers and our material of *C. applanatus* and *C. malachius* is from the wood of deciduous trees. There are only two other European species supposedly preferring substrates of conifers, *C. kubickae* and *C. stenocystis* [43,44]. Only *C. stenocystis* has a similar basidiospore shape to our new species, but it differs by its larger spores and base-inflated (mostly utriform) cheilocystidia and terminal cells in pileipellis [12]. Previous reports of *C. applanatus* from coniferous substrates, e.g., [3], may represent *C. pini* and need a revision. A search of publicly available sequences did not result in a positive match with our new species, and we believe that the diversity of recently described *Crepidotus* species is covered by our sequence sampling and is proof that *C. pini* has not been described recently. Among older species described outside Europe, *C. avellaneus* Hesler & A.H. Sm. and *C. campylus* Hesler & A.H. Sm. from the USA are similar, but they both grow on deciduous substrates and the first differs by its considerably small (up to 15 mm), white basidiomata and the second has cylindrical and capitate cheilocystidia [2].

The first phylogenetic studies with sufficient sampling across the genus *Crepidotus* were based on the LSU nr DNA region [45]. Subsequent *Crepidotus* studies used this genetic marker to assess the species delimitation and to estimate the interspecies relationships, e.g., [21,42,46]. With the general implementation of ITS as a universal fungal barcode [47,48], there is an increasing number of *Crepidotus* sequences available from mycological studies that are not aimed at the genus taxonomy. For this reason, some recent studies analyzed the ITS region in addition to the LSU or were used as the only genetic marker [12,20,49,50,51]. Since sequence data for many important *Crepidotus* collections are available only for one of these two regions of ribosomal DNA, we did not combine them in a single phylogenetic analysis.

## 5. Conclusions

This study solved the long-lasting question about the identity and delimitation of common globose-spored *Crepidotus* species, *C. applanatus* and *C. malachius*. Our phylogenetic study supported two species and placed them in two well-supported lineages within the genus *Crepidotus*. Their field aspect and microscopy are similar except for the spore size that we propose as a distinguishing character. Collections from coniferous substrate initially named as *C. applanatus* represent a third species, *C. pini*, which may also be identified based on the yellowish colours on its pileus surfaces when young, but a combination of spore and cheilocystidia characters are needed to recognise it from other similar species.

We highlighted that the majority of *Crepidotus* studies used either ITS or LSU nr DNA regions to support their species hypotheses. This makes the available sequence data inconsistent and did not allow us to combine both regions in a single two-loci tree. We applied a coalescent approach and analysed the ITS and LSU datasets separately, which also strengthened our argumentation that similar morphotypes represented by *C. applanatus* and *C. malachius* evolved convergently (which was supported by both analyses). The grouping of species with globose spores was not consistent between the trees. While ITS analysis supported the monophyly of the group, an LSU tree placed a small group of *C. ehrendorferi* and related globose-spored species close to species with non-globose spores. This clearly demonstrated that building a multi-locus tree including low copy genes is necessary to resolve phylogenetic relationships within the genus *Crepidotus*.

*Crepidotus pini* seems to be specialized to coniferous substrates and we did not find any apparent differences in the ecology of *C. applanatus* and *C. malachius*. The challenging question is if ecological conditions had driven the convergent evolution of this morphotype and if there is some niche differences and specialization between the two species dwelling deciduous substrates.

## Figures and Tables

**Figure 1 jof-08-00489-f001:**
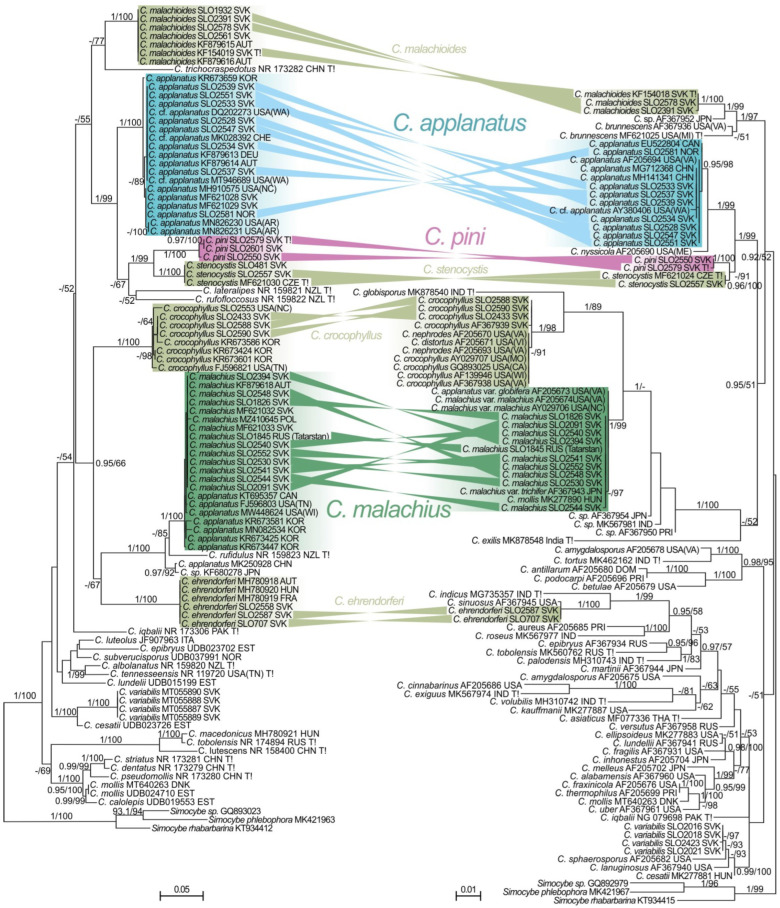
Maximum Likelihood (ML) trees based on phylogenetic analyses of nrITS (**left**) and nrLSU (**right**) DNA regions of the genus *Crepidotus*. Three *Simocybe* samples were used as an outgroup. Sequences retrieved from GenBank are provided with their original identifications, accession numbers, and codes of countries of the origin (https://www.iban.com/country-codes, accessed on 14 April 2022) and US and Canada state/province codes (https://www.fs.fed.us/database/feis/format.html, accessed on 14 April 2022). Sequences generated in this study are provided by herbarium numbers instead of Genbank accession numbers. Ex-type sequences are labeled with “T!”. ML bootstrap support values greater than 50% and BI posterior probabilities greater or equal than 0.95 are indicated at the nodes.

**Figure 2 jof-08-00489-f002:**
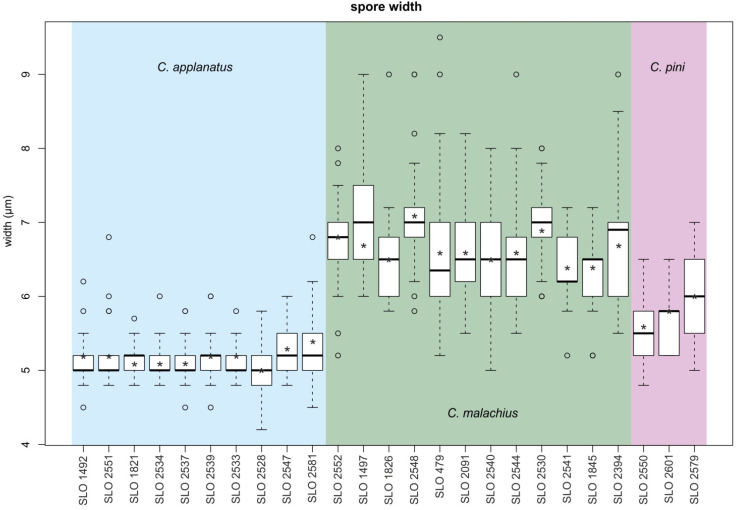
Boxplot charts comparing basidiospore width of three studied *Crepidotus* species. Asterisk represent average, thick horizontal line median and circles are outliers.

**Figure 3 jof-08-00489-f003:**
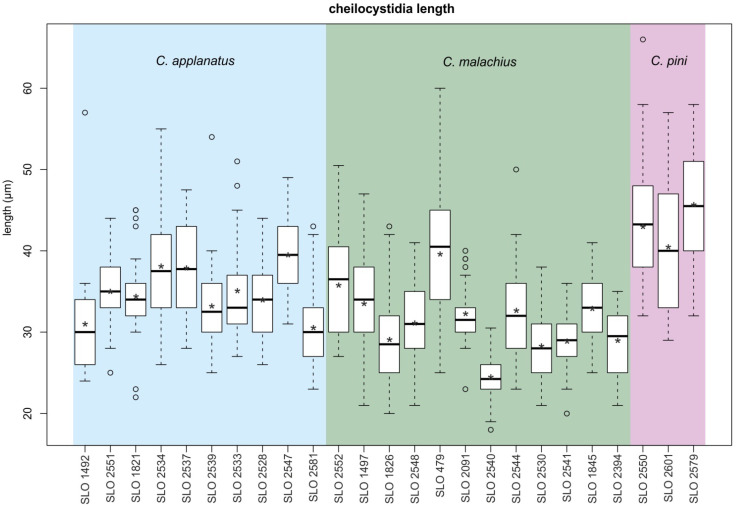
Boxplot charts comparing cheilocystidia length of three studied *Crepidotus* species. Asterisk represent average, thick horizontal line median and circles are outliers.

**Figure 4 jof-08-00489-f004:**
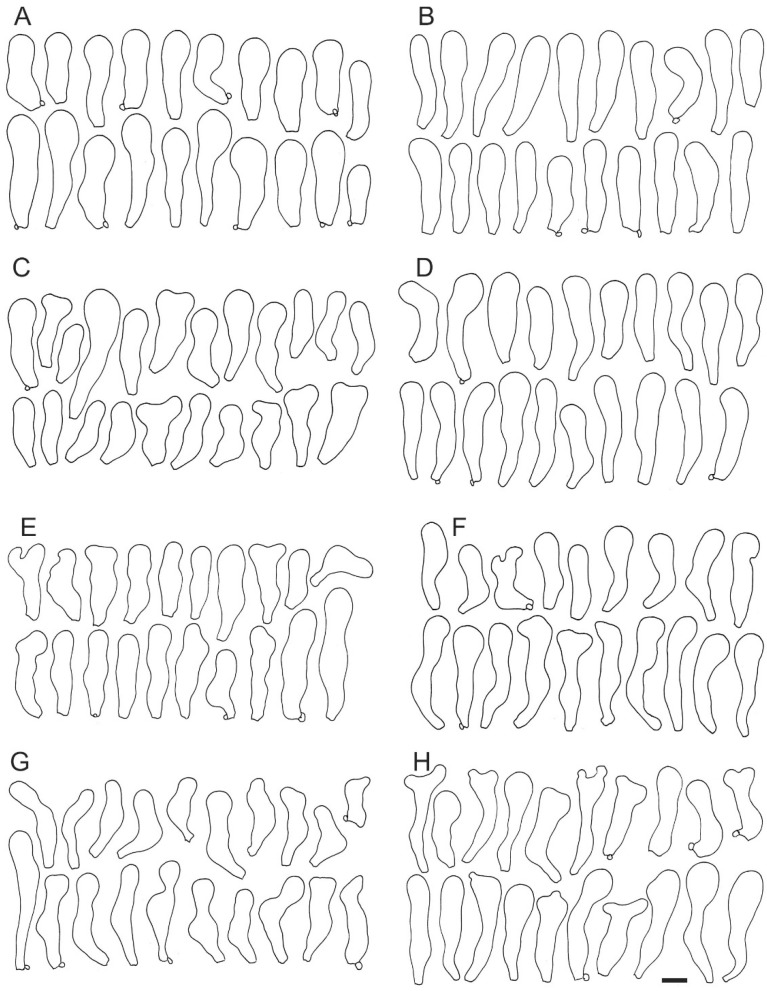
Cheilocystidia variability of *Crepidotus applanatus*. (**A**) SLO 1821, (**B**) SLO 2528, (**C**) SLO 1492, (**D**) SLO 2547, (**E**) SLO 2533, (**F**) SLO 2534, (**G**) SLO 2539, (**H**) SLO 2537. Scale bar = 10× μm.

**Figure 5 jof-08-00489-f005:**
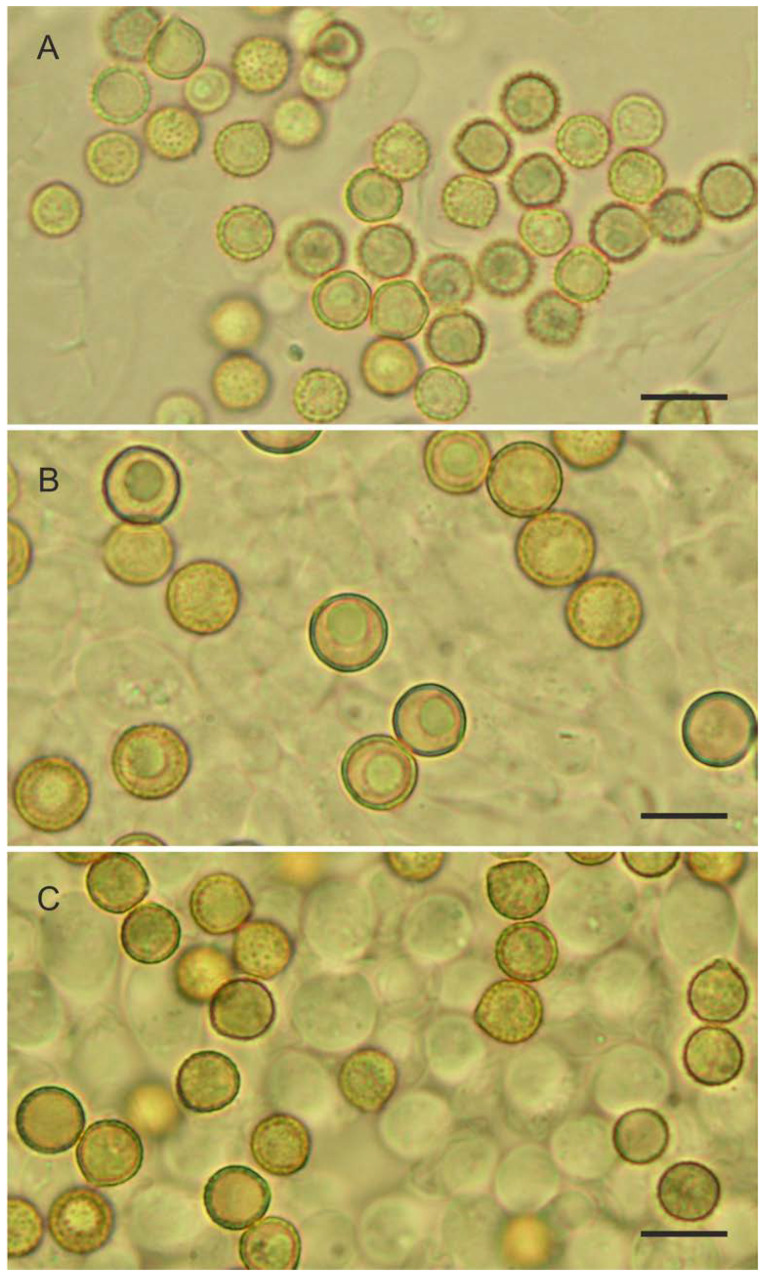
Light microscopic photos of basidiospores. (**A**) *Crepidotus applanatus* (SLO 2573), (**B**) *C. malachius* (SLO 479), (**C**) *C. pini* (SLO 2579). Scale bar = 10 μm.

**Figure 6 jof-08-00489-f006:**
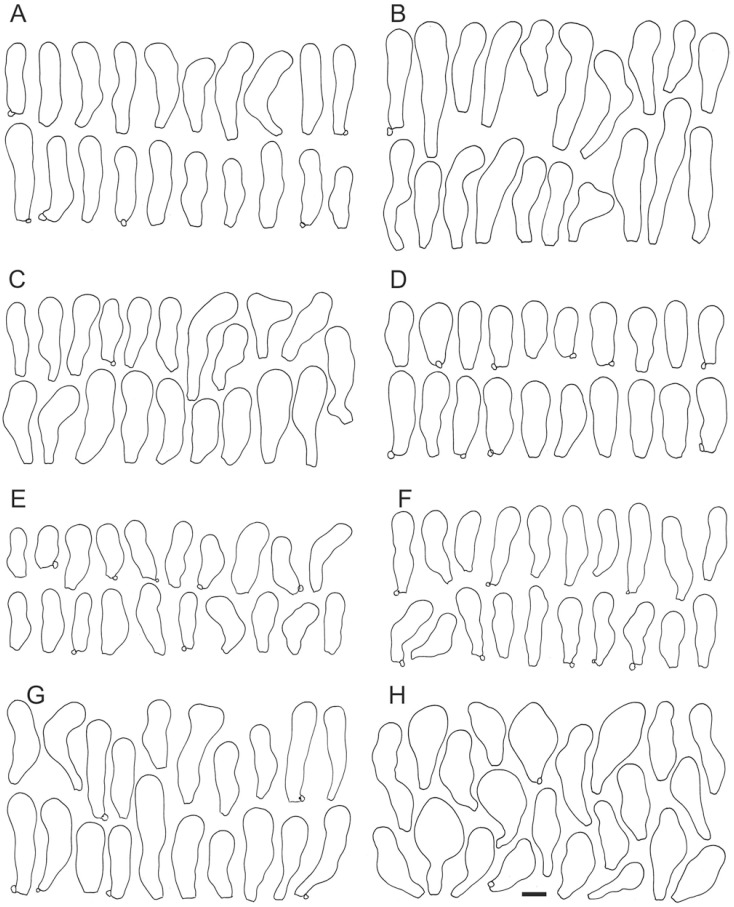
Cheilocystidia variability of *Crepidotus malachius*. (**A**) SLO 1845, (**B**) SLO 479, (**C**) SLO 1497, (**D**) SLO 2530, (**E**) SLO 2540, (**F**) SLO 2541, (**G**) SLO 2552, (**H**) SLO 2548. Scale bar = 10 μm.

**Figure 7 jof-08-00489-f007:**
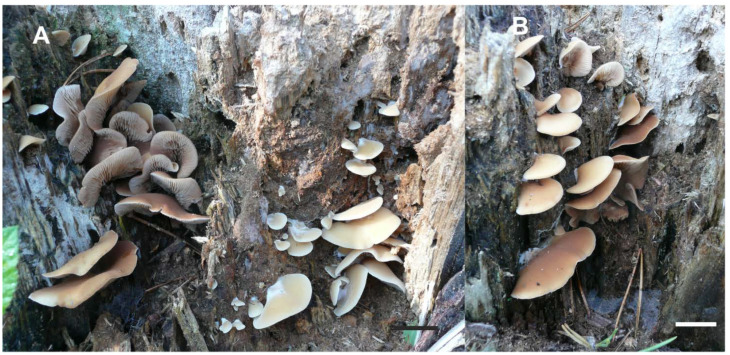
Sessile basidiomata of *Crepidotus pini* (SLO 2579). (**A**) Young basidiomata with greyish-yellow pileus surface (right) and mature basidiomata with brown lamellae (left), (**B**) mature basidiomata with more brownish pileus surface, (Scale bar = 1 cm).

**Figure 8 jof-08-00489-f008:**
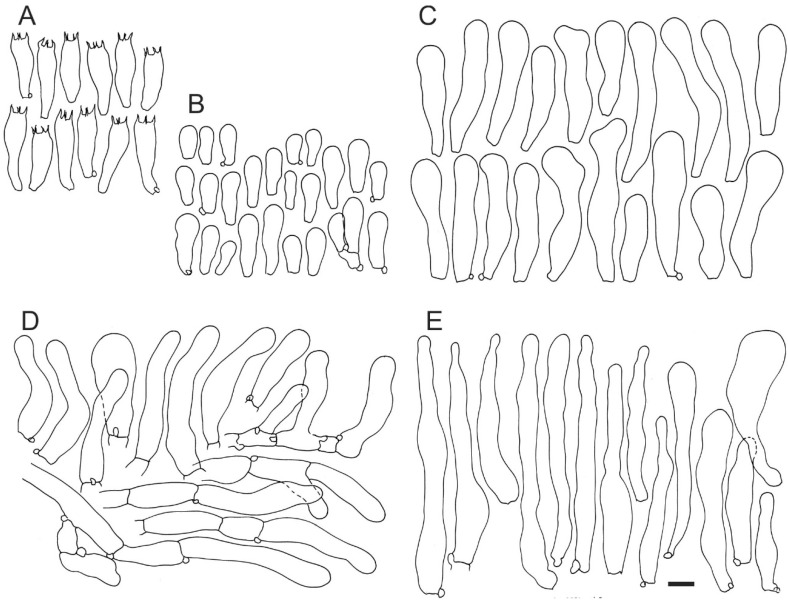
Microscopic elements of *Crepidopus pini* (SLO 2579). (**A**) basidia, (**B**) basidiola, (**C**) cheilocystidia, and (**D**,**E**) terminal elements in pileipellis. Scale bar = 10 μm.

**Table 1 jof-08-00489-t001:** Basidiospore and cheilocystidia characteristics measured on *Crepidotus applanatus* (*n* = 300), *C. malachius* (*n* = 360) and *C. pini* (*n* = 90). The size of basidiospores and cheilocystidia, the length/width ratio of basidiospores (Q), and the frequency of occurrence of cheilocystidia types (mostly = 50.1–99%, often = 25.1–50%, sometimes = 5.1–25%, rarely = 0.1–5%) was calculated based on the sequenced specimens (Appendix A); *n* = total number of measurements (30 measurements per specimen); the measurements are presented as averages ± standard deviation, values in parenthesis = minimum and maximum.

Characters	Species	Measurements
basidiospore size (μm)	*C. applanatus*	(4.2)4.8–5.2–5.5(6.8) × (4.2)4.8–5.1–5.5(6.8)
	*C. malachius*	(5)5.9–6.7–7.4(9.5) × (5)5.9–6.6–7.3(9.5)
	*C. pini*	(4.8)5.3–5.8–6.3(7) × (4.8)5.2–5.7–6.2(7)
basidiospore Q	*C. applanatus*	(0.94)0.99–1.00–1.02(1.10)
	*C. malachius*	(0.91)0.99–1.00–1.02(1.06)
	*C. pini*	(0.95)0.98–1.01–1.04(1.15)
cheilocystidia size (μm)	*C. applanatus*	(22)28.8–34.9–40.9(57) × (6)8.1–10–11.9(16)
	*C. malachius*	(18)25.1–31.5–37.8(60) × (7)8.2–10.1–11.9(19)
	*C. pini*	(29)35.5–43.4–51.3(66) × (7)8.5–9.8–11.1(14)
cheilocystidia type frequency	*C. applanatus*	mostly clavate; sometimes narrowly clavate; rarely utriform, ventricose, narrowly ventricose, narrowly cylindrical, forked, narrowly forked, lobate, narrowly lobate
	*C. malachius*	mostly clavate; sometimes narrowly clavate, ventricose; rarely utriform, narrowly utriform, narrowly ventricose, cylindrical, narrowly cylindrical, forked, narrowly forked, lobate, pyriform, globose
	*C. pini*	mostly narrowly clavate; often clavate; sometimes narrowly utriform; rarely utriform, narrowly cylindrical, forked, narrowly forked, lobate, narrowly lobate

**Table 2 jof-08-00489-t002:** Basidiospore size (μm) of *Crepidotus applanatus* and *C. malachius* in the selected monographic studies on the genus *Crepidotus*.

Source	*C. applanatus*	*C. malachius*
Pilát [1]	5–6.5	treated as a synonym of *C. applanatus*
Hesler & Smith [2]	4–5(5.5)	5–7.5(8.5)
Singer [41]	4.5–6	5.5–8.2; treated as a synonym of *C. nephrodes*
Watling & Gregory [38]	4.5–6 × 4.2–5.2(5.5)	-
Norstein [15]	5–7.5(8) × (4.5)5–7(7.5)	-
Senn-Irlet [3]	4.5–7 × 4.5–6.5	-
Bandala et al. [14]	(4.5)5.5–7(8) × (4.5)5.5–7(8)	treated as a synonym of *C. applanatus*
Consiglo & Setti [10]	4.9–5.8 × 4.7–5.4 (neotype 5–5.5 × 4.8–5.2)	6.2–7.3 × 5.8–6.9 (isotype 6.3–7.4 × 6–6.9)
Hausknecht & Krisai-Greilhuber [4]	4.5–6 × 4.5–5.5	6–8.5 × 5.5–7.5
Senn-Irlet [39]	4.5–7 × 4.5–6.5	-
Salaš [37]	5–5.5 × 4.5–5	6–8 × 6–8
This study	(4.2)4.8–5.5(6.8) × (4.2)4.8–5.5(6.8)	(5)5.9–7.4(9.5) × (5)5.9–7.3(9.5)

## Data Availability

Not applicable.

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
