# Peer review of "Phylogeny of Crepidotus applanatus Look-Alikes Reveals a Convergent Morphology Evolution and a New Species C. pini"

_jof, 2022, doi:10.3390/jof8050489_

Round 1
Reviewer 1 Report
This manuscript evaluates fungal specimens that have broadly been called " Crepidotus applanatus", and carefully compares the morphological differences between C. applanatus and it related species. The spore and cheilocystidia characters are confirmed as the key characteristics to distinguish these species, and one new species is described. However, this manuscript needs to be improved by English and some small problems. Some small problems are indicated in PDF with comments. Please see the attachment.

Author Response
Authors response:
page 1: we accepted all changes and we deleted “e.g.“ as suggested by the reviewer
page 2: all proposed changes are accepted
page 3: all proposed changes are accepted. we modified the first paragraph of 2.4 Morphological analyses and we expain which collections were used instance of which were excluded

Reviewer 2 Report
A very interesting, complete and well-illustrated manuscript of this group of Crepidotus species; however, it has an underlying problem because it is not clear that the group under study is monophyletic. Therefore, you have two options: 1) start from a monophyletic group clearly found by other authors (not citations 11 and 12, where the sample was also biased to say the group is monophyletic), or 2) use here a wider sample to clearly establish the monophyletic groups and work from them.
In the Introduction you established: “section Sphaerula…, serie Applanatus …. is defined by verrucose, globose to subglobose spores and white or whitish, hygrophanous and smooth pileus. Four species with the above mentioned characteristics are recognised in Europe: C. applanatus..., C. malachius..., C. malachioides… and C. stenocystis…”. However, you included other species in your phylogenetic analyses without specifying why, which resulted in your study group being paraphyletic, making it clear that you have to expand the group studied, to make sure that you are starting from a monophyletic group.
Supporting this, under Discussion you said: “Members of serie Applanatus defined by white or whitish and hygrophanous pileus surface and globose to subglobose spores are not monophyletic and represent a morphotype that evolved convergently.”, so I insist, you should have to include more species, to be sure that you are analysing a monophyletic group.
Your new species has not white or whitish pileus, another indication that your sampling must be expanded.
You should make a clear morphological comparison between C. pini and C. stenocystis as they are sister species (according to your phylogenetic analysis) and both grow on pine wood. You only compared, from the morphological point of view C. pini with C. applanatus and C. malachius.
It is not clear whether you reviewed the type specimens you mention in the manuscript, and other types of this group of species should have been checked.
A list of specific recommendations and corrections in the Supplementary Tables are attached.

Author Response
thank you very much for this deep and valuable review.
Our responses are in the attached file.

Reviewer 3 Report
Dear authors:
This paper is focusing on Phylogeny of Crepidotus applanatus look-alikes reveals a convergent morphology evolution and including a new species C. pini, which is interesting, but it can be accapted after major revision.
Some comments in the revised manuscript, please check it.
Comments:
1 Introduction: add the currently accepted taxa in 1)Mycobank and 2)Index fungorum, and 3)references;
2 added latest morpholgical and molecular references;
3 Phylogenetic Analyses: add the MP and Bayesin analyses to support you study
4 the nLSU is not good for the inner genus, so the author should build a new tree with nLSU for the family level among the genera related to Crepidotus; the ITS tree is not comprise enough taxa of this genus, so add more related species, in index fungorum presence of 568 records, so please check and add the related species; the author have to mark the TYPE specimen sequences, which is very important to your study for supporting your result, or this study can not give the conclusion " Our phylogeny did not support the close relationship of these two morphologically similar species and the grouping of collections labelled by both names within each phylogenetic species reflects unreliable species delimitations in the traditional literature."
5 add a paragraph for examined specimens from different country, and it has to include the type specimen
6 add a key for this genus worldwide
Kind Regards,

Author Response
thank you very much for your valuable comments
our responses are in the attached file

Round 2
Reviewer 2 Report
Dear Authors,
Your manuscript has improved, although some details are missing, which I describe in the attached file.
Best regards

Author Response
Thank you for the critical comments on our manuscript. We adopted or at least partly implemented nearly all of them with exception of the key. MP analysis and expading of LSU to the family rank are explained. Upon facing possible rejection from the editors we will do the changes. We appreciate opinion of the reviever but in these three points we think that this is matter of opinion more than scientific relevance and quality of the manuscript.
We are very gratefull for these valuable comments, we accepted all of them including the addition of Conclusion."

Reviewer 3 Report
Dear authors:
This is a bad attitude to reply to reviewers comments and downstreaming rigorous scientific way to do youself manuscript.
1) In introduction: "More than 200 species are described worldwide [6]", in which the reference was published 2008, and it was 14 years ago, and the author want to publish a new species, so you have to check the latest papers (I gave you comments last time as " Introduction: add the currently accepted taxa in 1)Mycobank and 2)Index fungorum, and 3)references;
", but you ignore it ).
2) the author DID NOT agree with my comments, but all of them are to improve your manuscript, and the author have to follow them in last time commments as following, or I am going to reject it.
The last time comments:
1) added latest morpholgical and molecular references;
2) Phylogenetic Analyses: add the MP and Bayesin analyses to support you study
3) the nLSU is not good for the inner genus, so the author should build a new tree with nLSU for the family level among the genera related to Crepidotus; the ITS tree is not comprise enough taxa of this genus, so add more related species, in index fungorum presence of 568 records, so please check and add the related species; the author have to mark the TYPE specimen sequences, which is very important to your study for supporting your result, or this study can not give the conclusion " Our phylogeny did not support the close relationship of these two morphologically similar species and the grouping of collections labelled by both names within each phylogenetic species reflects unreliable species delimitations in the traditional literature."
4) add a key for this genus worldwide
5) in this paper, it proposed a new species as C. pini, why is it a new species? you have to check all of the taxa of this genus, perhaps other mycologist had published it or reported it.
Kind Regards
Author Response
Thank you again for valuable comments. WE appreciate all of them and we adopted nearly all at least partially. We did a lot of effort to improve the manuscript, based on the reviewers comments, sampling in both trees were expanded, available types were added, BI analysis was performed, comments on C. pini were provided. We appreciate opinion of the reviever but weith all respect we think that some comments are beyond scope of our study, apthogh we agree they can improve impact of the paper (global Crepidotus key).

Round 3
Reviewer 3 Report
Dear Author,
Some commens follow and check them.
1) In Table S2, many sequences are unpublished, which indicates some of them were bad sequences, so the alone species in the tree has to delete;
2) for the tree, 3) the nLSU is not good for the inner genus, so the author should build a new tree with nLSU for the family level among the genera related to Crepidotus; the ITS tree is not comprise enough taxa of this genus, so add more related species, in index fungorum presence of 568 records, so please check and add the related species;
3) in the molecular tree, the C. sp. (367952) and C. ver. (367943) do not work and disrubt the topology of this tree , to delete them in the new tree;
3) in the preset paper, a new species is found, so add a key for this genus worldwide;
All the best,
Author Response
Many thanks again to reviewer 3 for the valuable comments.
We appreciate the opinion of the reviewer but we do not share it and we do not wish to make the proposed changes. I believe that we make a clear explanation that will satisfy both reviewer and editors.
